# Research on High-Frequency Information-Transmission Method of Smart Grid Based on CNN-LSTM Model

Xin Chen [1,2]

1 College of Electrical Engineering and New Energy, China Three Gorges University, Yichang 443002, China; cx220612@163.com
2 Hubei Provincial Key Laboratory for Operation and Control of Cascade Hydropower Station, China Three Gorges University, Yichang 443002, China

**Abstract:** In order to solve the problem of the slow transmission rate of high-frequency information in smart grid and improve the efficiency of information transmission, a research method of high-frequency information transmission in smart grids based on the CNN-LSTM model is proposed. It effectively combines the superiority of the CNN algorithm for high-frequency information feature extraction and the learning ability of the LSTM algorithm for global features of high-frequency information. Meanwhile, the client buffer is divided by the VLAN area division method, which avoids the buffer being too large due to line congestion. The intelligent control module is adopted to change the traditional control concept. In addition, the neural network optimization control module is used for intelligent control, which ensures the feedback speed of the control terminal and avoids the problem of increasing the buffer area caused by the feedback time difference. The experimental results show that via the method in this paper, the total efficiency of single-channel transmission reaches 96% and the transmission rate reaches 46 bit/s; the total efficiency of multiplex transmission is 89% and the transmission rate reaches 75 bit/s. It is verified that the method proposed in this paper has a fast transmission rate and high efficiency.

**Keywords:** CNN-LSTM; smart grid; high-frequency information; single transmission; multiple transmission

## 1. Introduction

With the pace of China's modernization, the energy crisis has become increasingly severe, and people's awareness of environmental protection has gradually increased. Smart grids have the advantages of high efficiency, cleanliness and saving energy [1]. The construction of smart grids has occupied a favorable development form, and has risen to the national strategic level [2]. A new type of power grid formed by the organic combination of physics, sensing and measurement technology, information technology, computer technology, control technology and communication technology is called the smart grid [3–6]. Communication not only undertakes the production command and dispatch of the power industry, but also provides services for the organization and management of the power industry and automatic communication, which play a supporting and guaranteeing role in smart grids [7–9].

The construction of smart grids combines the current situation of China's electric power communication development, and the communication network construction of the state grid has made certain achievements [10]. However, no matter the smart distribution network or the electricity-consumption information collection, the difficulty in the construction of smart grids is the construction of communication system [11–14]. The reliability of the communication system determines the quality of the whole system to some extent, and communication is the necessary means of distribution automation [15]. Carrying out the mission where the commands of the control center are sent to all executive agencies or remote terminals, and the information collected by the remote monitoring

unit is transmitted to the control center, are the key technologies of building an intelligent distribution system [16–19].

In order to reduce the amount of operational information transmission and improve the speed of operational information transmission, many researchers have made bold research. However, most of them are for single-layer transmission, which are difficult to improve the speed of information transmission [20–22]. With the rapid development of computer network technology, the high-frequency information-transmission mode of smart grids has become a hot spot in the research field of power systems [23]. This operation mode effectively improves the power operation cycle, makes full use of resources and reduces the development cost, but involves a huge amount of information contained in computer assistance [24,25]. Therefore, the transmission of high-frequency information that only adopts a single-layer transmission mode in power-grid operation will take up a lot of time, bring difficulties to the real-time operation of the power grid and fail to meet people's massive demand for power.

The authors of [26] state that under the smart development trend of the digitization of the secondary circuit, abnormal data problems in data-transmission networks such as network storms also bring certain hidden dangers to the safe and stable operation of secondary equipment while bringing convenience to data sharing. To this end, it analyzes the main problems affecting the reliability of data transmission, and proposes a packet-management technology based on data-flow control, which effectively improves the reliability of data transmission. At the same time, a solution based on cross-network port-flow control was proposed for the problem of cross-network data exchange. The authors of [27] discuss the basic transmission scheme of power-dispatching data, clarify the basic principles of power-dispatching-data network design, and on this basis, comprehensively expound the specific design of power-dispatching-data network security protection. The purpose of this discussion and analysis is to ensure the safety of power-dispatching-data network transmission and lay a solid foundation for the long-term development of the power system. The author of [28] explores how the traditional wireless power transmission-system secondary side adopts an uncontrolled rectifier circuit, which can only control the phase shift angle of primary side. In view of this, a wireless power and information transmission system with a chopper circuit as the secondary side is proposed. Specifically, a controllable device and a diode are added between the load and the uncontrolled bridge on the secondary side, so that the bilateral phase-angle control is achieved and a constant output load voltage is achieved via the feedback control of the secondary side. Meanwhile, based on the structure of the receiver side of the uncontrolled bridge on the secondary side, the information modulation is designed by changing the switching frequency of the fully controlled device on the receiver side. The author of [29] proposes an automatic verification method for the security of user information transmission in open public networks based on media-access-control security-encryption transmission. To be concrete, an open public-network user-information-transmission channel model is constructed, several key management and identity authentication methods under the IEEE802.1X standard protocol are adopted to carry out encryption design for open public-network user-information transmission, and the digital certificate identification protocol is applied to construct user-information-transmission security automation. The verification model adopts the medium-access-control security-encryption-transmission method to realize the encryption verification and certificate authentication of the user-information output. The authors of [30] deem that OTN technology plays a crucial role in solving the bottleneck problem of power communication transmission. It mainly analyzes the application of OTN technology from the three aspects of networking strategy, equipment model selection and system capacity, and expounds its use situation and use effect to better introduce the current use status and use requirements of the power communication-transmission network. The emerging new technologies have also brought new challenges to the power information communication-transmission industry in terms of transmission efficiency. At this stage, in order to effectively improve the efficiency of power information

communication transmission and ensure its quality, it is necessary to make use of the important achievements in the development of modern scientific information technology.

Aiming at the above problems, this paper conducts research on the high-frequency information-transmission method of smart grids based on CNN-LSTM model. This algorithm combines the feature-extraction ability of CNN to data information, makes full use of LSTM's memory ability to learn the global characteristics of a high-frequency data platform and reduces the amount of text data. This method can effectively reduce the client buffer of a smart grid platform, reduce the complexity of the system and facilitate long-distance transmission.

## 2. CNN-LSTM Model

The continuous development of society has promoted the progress of information technology to a certain extent, and the traditional communication mode can no longer meet the development of the country. Therefore, to continuously develop smart grid telecommunication technology, we should not only apply it to real life but also integrate advanced telecommunication-related science and technology into relevant construction. It is of positive significance to build power information communication technology in the era of the smart grid, and the application of power information communication technology can affect the development of smart grids to a certain extent.

The CNN model is one of the most popular and widely used models in the field of deep learning in recent years. It adopts the method of local connection and weight sharing, directly obtaining effective representation from the original data by alternately using a convolution layer and pool layer, and automatically extracting local features of data. Meanwhile, it establishes dense and complete feature vectors. However, we only pay attention to the local features, but not the global features, which are prone to the phenomenon of multiple types of information and affect the results of information classification.

LSTM is a serial processing method that can contact the whole information well and classify the data information accurately. However, due to its complex structure, with the increase in the input data, the amount of the calculation also increases, thus reducing the connection between the whole information and the accuracy of the algorithm. Pursuant to the above, in this paper, combining the respective advantages of the CNN algorithm and the LSTM algorithm, a CNN-LSTM model is formed to train the data set, and the high-frequency information transmission method of the smart grid is studied through the trained model.

The structure of the CNN-LSTM model is shown in Figure 1. It mainly includes the input layer, CNN layer, LSTM layer, dropout layer, and output layer. What is noteworthy is that the output of the previous layer is used as the input of the next layer.

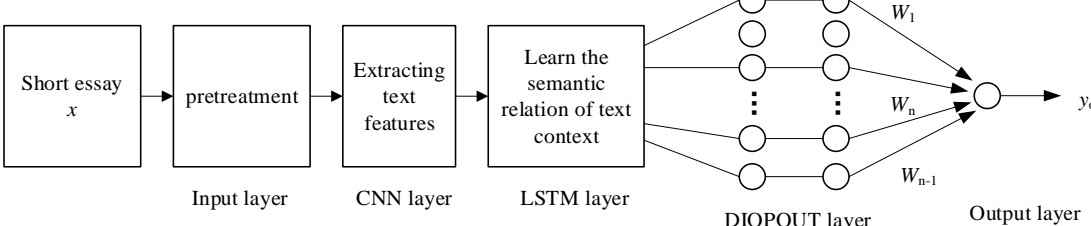

**Figure 1.** CNN-LSTM model structure.

As can be seen from Figure 1, at the input layer, the high-frequency information is first preprocessed and the word2vec tool is used to convert the $m$ information of the high-frequency data into the word vector processing of the $n$-dimensional vector and generate a converted $m \times n$ data matrix. Then, the grayscale-value image is input to the CNN layer. In the CNN layer, a convolution kernel of size A is used to extract the feature value image and input it to the LSTM layer. The LSTM layer will learn contextual semantics in the form of information feature words and pre- and postsequences through the input gate, forget

gate and output gate, and finally output a vector. This vector is fed into the dropout layer, which ignores part of the feature detector in a random way (letting part of the hidden layer nodes have a value of 0). Finally, the output value of the fully connected layer is sent to a Softmax classifier for classification so as to determine the probability of high-frequency information labels. The calculation formula is:

$$h_w(\boldsymbol{y}) = \left(1 + \exp\left(-\boldsymbol{w}^{\mathrm{T}}\boldsymbol{y}\right)\right)^{-1} \tag{1}$$

Among them, $\boldsymbol{w} = \begin{bmatrix} w_1 & w_2 & \cdots & w_{n+} \end{bmatrix}^{\mathrm{T}}$ describes the model training parameters; $\boldsymbol{y} = \begin{bmatrix} y_1 & y_2 & \cdots & y_{n+1} \end{bmatrix}^{\mathrm{T}}$ describes the output of multiple neurons; and $h_w(\boldsymbol{y})$ describes the probability of belonging to a certain class via judging.

## 3. Smart Grid High-Frequency Information-Transmission Method

Smart power consumption is an important part of the smart grid, and also the hub connecting the power supply department and users. The social communication industry slope is large. The electricity-consumption information-collection system is the technical realization of intelligent electricity consumption. It is a complete network and system used to measure, collect, store, analyze and use user electricity-consumption information. The establishment of the electricity information-collection system will change the current situation of one-way flow of electricity and information flow at the bottom of the street, and provide a platform and technical support for a two-way comprehensive interaction between users and the power grid. Through the information interaction with the power grid, users will know the load situation and electricity price information of the power grid at any time, so as to actively participate in the operation of the power grid.

### 3.1. Smart Grid Platform Buffer Division

In order to apply VLAN technology to the intelligent control system of the high-frequency information-transmission rate of smart grids, a setting module is added to the system to facilitate the division of data areas, and at the same time to ensure that the normal transmission of information is not affected during the process of division. The division of the buffer area of the smart grid platform is achieved through data classification, and data preprocessing and attribute source determination are also required. The classification of buffer data in the intelligent control system of the grid platform can be divided according to the source as follows:

$$i = \frac{E - U}{Z} = \frac{EZ_1Z_2 + E_iZ_1Z_2 - EZ_2Z_i}{Z(Z_1Z_2 + Z_1Z_i + Z_2Z_i)} \tag{2}$$

In the formula, $i$ represents the actual divided area; $E$ represents the type of the area; and $Z, Z_1, Z_2, Z_i$ represent different types of data, each of which comes from the same source. Before data filling, some data preprocessing is required, so as to facilitate the system to perform identification, which is as follows:

$$E = U + \frac{R}{U}(P - P_i) + \frac{X}{U}(Q - Q_i) + \frac{R}{U}P + \frac{X}{U}Q \tag{3}$$

In the formula, $R$ represents the allocation type of buffer data; $X$ represents the data-transmission attribute; and $Q$ represents the identification attribute of the e-commerce platform.

Assuming that the actual size of the buffer of the e-commerce platform is $B$, after the data is filled, the buffer data bits change from $D$ to $D_i$; that is, $\Delta T = D + 1 - D_i$. Again, assuming that the buffering speeds of the used data are $T$ and $T_i$, respectively, there are:

$$B_i - B = (R - R_i) \times (T - T_i) \tag{4}$$

$$B_i = 0, B = (R - R_i) \times (D + \Delta T + D_i) \tag{5}$$

$$\boldsymbol{W} \geqslant (R - R_i) \times (D + \Delta T + D_i) \tag{6}$$

In the above formulas, the value of client buffer $B$ and $B_i$ is a fixed value, and usually, it is necessary to consider relevant factors to ensure the reduction in the buffer [18]. The buffer that appears in the way of feedback delay is reduced by the exchange of the control system. Additionally, the transmission rate of the buffer data of the e-commerce platform is limited, and $\Delta T$ is used to indicate the usage of the associated data, so as to achieve the purpose of VLAN division and decomposition.

### 3.2. Optimizing Control System Using Neural Network Control Module

Using the neural network control module to optimize the e-commerce system is to increase the feedback speed of information and avoid data reservation in the information-filling process of the roadway. The optimization process can be described by the following equation:

$$P_i = \mathrm{MAX}\left\{ P, P_i^D \right\} \tag{7}$$

In Equation (7), $P_i^D$ represents the partitioned buffer saturation value. In order to obtain the relationship between the saturation data and the data-transmission reservation ratio, the saturation value is simplified as follows:

$$P_i^D = C_p^i / C_{\mathrm{max}}^i \tag{8}$$

Among them, $C_{\mathrm{max}}^i$ describes the contained information space, and $C_p^i$ describes the collection-information space. Increasing the feedback speed will reduce the buffer of the smart grid platform. The expression is:

$$p = \boldsymbol{W}_c / \boldsymbol{W}_B \tag{9}$$

The sizes of $\boldsymbol{W}_B$ and $\boldsymbol{W}_C$ determine the size of the buffer used by the process. In the process of channel reservation calculation, the premise of data storage is to calculate the reserved space, as follows:

$$G = \sum_{r=1}^{t} \sum_{q=1}^{k_2} \left\| \boldsymbol{W}_i^{\mathrm{T}} x_{ir} - \boldsymbol{W}_i^{\mathrm{T}} x_{irq} \right\|^2 B_{irq} \tag{10}$$

In the formula, $G$ represents the storage capacity of the space, and the order of the smart grid platform is guaranteed by the reserved calculation of the space, which can also save the buffer space in the next part. Finally, via model proposed in this paper, the intelligent control client buffer of the high-frequency information transmission rate of the smart grid platform can be expressed as:

$$H_2 = \sum_{r=1}^{t} \sum_{q=1}^{k_2} \left( x_{ir} - x_{irq} \right) \left( x_{ir} - x_{irq} \right)^{\mathrm{T}} B_{irq} \tag{11}$$

It can be seen from the formula that by using VLAN technology to divide and fill the e-commerce platform buffer, the reserved redundant area is transferred to the data, which reduces the buffer area and improves the data transmission speed. Meanwhile, the control system is optimized by using the neural network control module, which greatly improves the feedback speed and reduces the buffer space of the e-commerce client.

### 3.3. Packing Method of Data-Transmission Information

The main body of the high-frequency information transmission of the smart grid under the transfer of big-data resources is the form of data with a large volume. Therefore, data transmission should be packaged in the corresponding PCI address mapping bytes.

According to the requirements of the communication PCI address mapping mechanism, each packing unit is controlled at 288 characters.

Improving the packaging method of high-frequency information in smart grids can effectively improve the security of wireless data communication. The packaging process of high-frequency data-transmission information is to determine the receiving task connection according to the operation of the initialization program and the loading of port information, and at the same time add the key and certificate, create a waiting buffer queue, set the corresponding waiting parameters, perform transmission according to the multiprocess network information, and finally integrate, create a thread pool, and prepare to pack the data. During the packaging process, the packaging is classified and packaged according to the data type, the degree of data confidentiality, and the purpose of data transmission. In order to ensure the integrity of the packaged data, the packaging compression ratio cannot exceed the requirements of Formula (12).

$$f = \lim_{t \to \infty} \left( \frac{j_n \cdot l_n}{t \cdot q} + z \right) \mathrm{d}x \tag{12}$$

where $f$ represents packing compression ratio, $t$ represents the data type, $q$ represents the degree of data confidentiality, and $z$ represents the integrity of the transmission link.

## 4. Simulation

In order to verify the effectiveness of the CNN-LSTM model proposed in this paper for the high-frequency information transmission method of smart grid, a comparative simulation experiment is designed. Meanwhile, the traditional method is simulated and compared with the proposed method in two cases of single-channel transmission and multiple-channel transmission, respectively. In two different cases, the two indicators of total transmission efficiency and average received bit rate are compared and analyzed.

### 4.1. Parameter Setting

Since the information flow is scalable during transmission, different receiving ends may have different security requirements for information flow during transmission. Hierarchical transmission of the identified data storage format not only saves bandwidth, but also effectively increases the speed of information transmission, which is beneficial to avoid network congestion.

The principle of information layering is to perform hierarchical coding on a digitized information flow of a certain resolution, and divide it into several levels according to the full refinement procedure. Each level component is transmitted by an independent information flow, and layer information can guarantee the minimum level of information quality requirements. To be concrete, the basic layer is composed of the quality information flow of all high-frequency information-transmission modules of the smart grid, and the flow rate of this information flow is about 125 kbit/s; the second layer is the additional information of the basic layer, and the communication-service quality-information flow can make the flow rate increase to 1.3 Mbit/s; the third layer of information flow is responsible for enhancing the transmission quality of the first two layers of modules, so that the flow rate rises to more than 4 Mbit/s. When carrying out high-frequency information transmission of a smart grid, it is necessary to provide a rate of 15 Mbit/s or even 25 Mbit/s, so that the receiver can receive data at different levels synchronously, and decode each layer of data. If there are many layers, the data will be reconstructed. The smaller the distortion, the higher the transmission security.

The polling feature parameter is introduced to judge the experiment. The experimental process parameters are as follows: the amount of information is 1 GB, the calling network layer is 3 layers, the node is GGSN, and the minimum frequency limit in the differentiation layer is not less than 15%. In order to ensure the validity of the experiment, it is necessary to randomly select the parameters of the experimental process to make the experiment more complete.

### 4.2. Experimental Setup

In this paper, SQL Server 2008 R2 is used as the database development platform, the functions of storing, processing, querying and managing the original data of the system are realized through LabVIEW instructions, and the experiment is supposed to be conducted in the conditions that 100 smart receivers are randomly distributed within a circular range of 200 m from the center of the power grid, the carrier frequency is 2.0 GHz, the frequency bandwidth is 15 MHz, the transmit power is 50 dBW and the noise interference factor is 10 dB.

In each transmission layer, the power grid center can obtain the signal-to-noise ratio of different users through the uplink channel and calculate the maximum modulation and coding level accepted by different users through the signal-to-noise ratio.

### 4.3. Unicast vs. Multiplex Rate Comparison

In the case of single-channel transmission, 100 smart receivers are all connected to the video of this channel, and at the same time the smart receivers are set to be randomly distributed, the simulation times are set to be 30, and the results are averaged.

Figure 2 shows the total transmission efficiency and average received bit rate curves of the two methods in single-channel and multichannel transmission modes when the number of experiments is 0 to 30.

From Figure 2a,b, it can be seen that under the same information-transmission speed, the total transmission efficiency and average received bit rate of the method proposed in this paper are both higher than those of the traditional method. For the CNN-LSTM model-based method proposed in this paper, when the information is 32 bits, the highest total transmission efficiency and average received bit rate is 99% and 180 kbps, respectively; when the information is 96 bits, the highest total transmission efficiency and average received bit rate is 96% and 171 kbps, respectively. In contrast, in the traditional method, when the information is 32 bits, the total transmission efficiency and the average received bit rate are up to 72% and 120 kbps, respectively; when the message is 96 bits, the highest total transmission efficiency and the highest average received bit rate are 69% and 100 kbps, respectively.

It can be seen from Figure 2c that for 32-bit information transmission, when the number of experiments is 5, the total transmission efficiency using traditional method is 30%, while the total transmission efficiency of the method in this paper is 80%. Using the traditional method, when the number of experiments is 30, the total transmission efficiency is 65%, while the total transmission efficiency of the method in this paper is 94%. For 96-bit information transmission, using the traditional method when the number of experiments is 30, the total transmission efficiency reaches a maximum of 60%, while the total transmission efficiency based on the method in this paper is 89%.

It can be seen from Figure 2d that under different transmission speeds, the average received code rate of the method in this paper is higher than that of the traditional method. When the information is 32 bits, the average received code rate of the method in this paper can reach up to 900 kbps; when the information is 96 bits, the average received bit rate of this method can reach up to 850 kbps.

According to the above comparison, the total transmission efficiency and average received bit rate of the method in this paper are higher than those of the traditional method. In this case, the transmission rates of the two methods are analyzed, and the results are shown in Table 1.

It can be seen from Table 1 that the single-channel transmission rate of the method based on this paper is obviously faster than the single-channel transmission rate of the traditional method, and when the number of experiments is 30 times and the amount of information is 32 bits, the transmission rate of the traditional method reaches 11 bit/s, while the transmission rate of the method in this paper is up to 24 bit/s; under the 96-bit information volume, the transmission rate of the traditional method is up to 24 bit/s, while the transmission rate of the method in this paper is up to 46 bit/s.

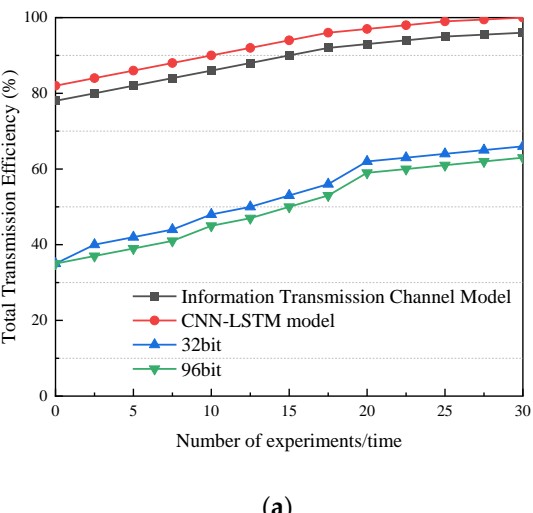

(**a**)

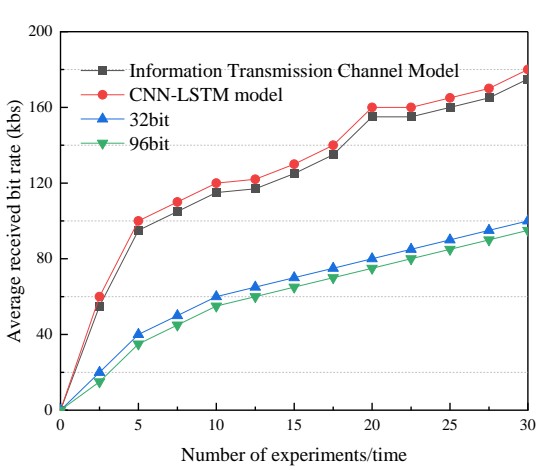

(**b**)

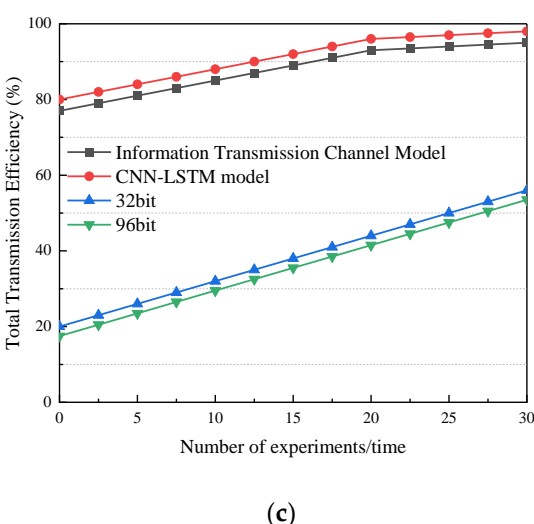

(**c**)

**Figure 2.** *Cont.*

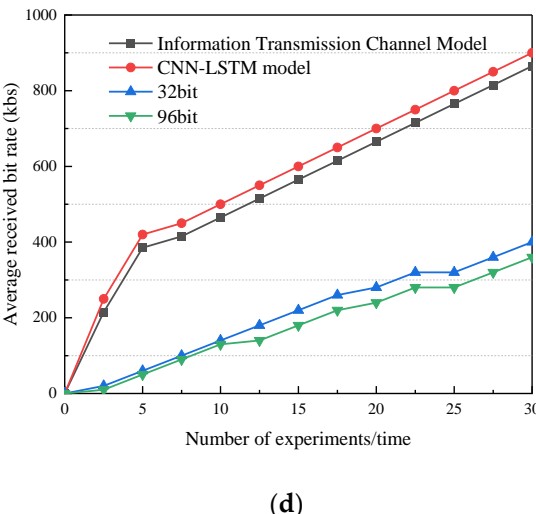

(**d**)

**Figure 2.** Comparison of different paths for transmission. (**a**) Overall efficiency of single-channel transmission; (**b**) average received bit rate of single-channel transmission; (**c**) overall efficiency of multiplexing; (**d**) average received bit rate of multiplex transmission.

**Table 1.** Single-channel transmission rate of two methods.

| Test Times/Time | Information Transmission Channel Model | | CNN-LSTM Model | |
|---|---|---|---|---|
| | 32 bit | 96 bit | 32 bit | 96 bit |
| 5 | 10 bit/s | 20 bit/s | 19 bit/s | 42 bit/s |
| 10 | 12 bit/s | 21 bit/s | 20 bit/s | 41 bit/s |
| 15 | 13 bit/s | 21 bit/s | 22 bit/s | 40 bit/s |
| 20 | 14 bit/s | 25 bit/s | 22 bit/s | 45 bit/s |
| 25 | 15 bit/s | 25 bit/s | 24 bit/s | 45 bit/s |
| 30 | 11 bit/s | 24 bit/s | 24 bit/s | 46 bit/s |

In the case of multiplexing, the transmission rates of the two methods are analyzed, and the results are shown in Table 2.

**Table 2.** Multiplexing rates of two methods.

| Test Times/Time | Information Transmission Channel Model | | CNN-LSTM Model | |
|---|---|---|---|---|
| | 32 bit | 96 bit | 32 bit | 96 bit |
| 5 | 10 bit/s | 20 bit/s | 19 bit/s | 42 bit/s |
| 10 | 12 bit/s | 21 bit/s | 20 bit/s | 41 bit/s |
| 15 | 13 bit/s | 21 bit/s | 22 bit/s | 40 bit/s |
| 20 | 14 bit/s | 25 bit/s | 22 bit/s | 45 bit/s |
| 25 | 15 bit/s | 25 bit/s | 24 bit/s | 45 bit/s |
| 30 | 11 bit/s | 24 bit/s | 24 bit/s | 46 bit/s |

It can be seen from Table 2 that the traditional method is used to compare and analyze the transmission rate in the case of multiple transmission, and it is found that the transmission rates of the two are basically unchanged when the amount of information is 32 bits. When the amount of information is 96 bits, the transmission rates of the two are both increased. However, the transmission rate of the method proposed in this paper increases significantly, and the fastest one reaches 75 bit/s.

## 5. Discussion

With the increasing complexity of the electric power communication network, the carrying services become more diverse and complicated, and the topological structure, communication equipment, networking technology, service types, fault protection and re-

covery methods of the electric power communication network have undergone tremendous changes. The diversified development of network-based services makes research on service transmission channels an important topic, especially research on service channel reliability. Next, considering the business importance, this paper re-examines the traditional research methods of business importance from the perspective of the concept and evaluation criteria of business importance.

## 6. Conclusions

With the comprehensive development of smart grid construction and the continuous progress of communication technology, the research of power-consumption information-acquisition systems is gradually becoming a hot spot. As the channel of data transmission in power information-acquisition systems, the choice of communication mode plays a vital role in the accurate, reliable and economical operation of the system. The research on the CNN-LSTM model-based high-frequency information-transmission method of smart grids proposed in this paper uses CNN to process multisource data and extract effective features, and at the same time uses LSTM networks to analyze time-series data. Its advantages are as follows:

(1) When the CNN-LSTM model algorithm transmits high-frequency information without adding any other artificial features, the total transmission efficiency is 89%, which not only effectively improves the security of data information but also obviously improves the processing speed. The classification effect of CNN-LSTM model algorithm is better than that of traditional algorithms.
(2) According to CNN-LSTM model, it can ensure the real-time data transmission and the stability of system operations, and contribute to the promotion of the smart grid.
(3) In the next step, we will study how to select a reasonable convolution kernel size in CNN layer for gray images with different sizes, so as to reduce the calculation amount of the algorithm.

**Funding:** This research received no external funding.

**Institutional Review Board Statement:** Not applicable.

**Informed Consent Statement:** Not applicable.

**Data Availability Statement:** The data used in this study are available from the corresponding author upon reasonable request.

**Conflicts of Interest:** The authors declare no conflict of interest.

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
