# Peer review of "Research on High-Frequency Information-Transmission Method of Smart Grid Based on CNN-LSTM Model"

_information, doi:10.3390/info13080375_

Round 1

Reviewer 1 Report

The paper presents the research on the high-frequency information transmission method of smart grid based on CNN-LSTM model. CNN is used for processing multi-source data and extracting effective features.  LSTM model is used for time series data analysis. Multiplexing rates are analyzed.

However, some points should be described in more detail.

The traditional method should be explained better as well as structure of CNN-LSTM.

Author Response

Thank you very much for your letter reminding me to rework the manuscript, and I am very sorry for the delay in rework. Since I have asked a professional translation agency to polish the language of the article, it took a long time to polish the article, so I have only completed the repair now. I would like to express my sincere apologies for the inconvenience caused to you. Please see the attachment.

Reviewer 2 Report

1.     In the Abstract, first provide some background knowledge and discuss the existing problems. Then, start with the discussion of the proposed work.

2.  There is no proper justification as to why CNN and LSTM have been used in the proposed work. Why not any other algorithms/methods? What are their advantages over other methods?

3.     The literature provided in the Introduction section comprises less number of articles. More closely related articles from the ongoing year, i.e., 2022, must be added to the text.

4.     The simulation parameters and the corresponding values provided in section 4.2 must be accompanied by strong justification as there is no research without justification. Moreover, the simulation tools must also be discussed.

5.     In Figures 2(a)-2(d), the comparison is made with the "traditional method" and "ways to improve". Using these terms makes the model and its simulations ambiguous. Therefore, it is advised that proper names should be used for the comparison techniques.

6.     In Figures 2(b) and 2(d), the y-axis represents the average received bit rate and it is mentioned in the axis title that the result is in %. However, the values range from 0-200 and 0-1000, respectively. As per basic knowledge, the percentage values cannot exceed 100. Therefore, both the figures need to be reconfigured.

7.     The authors are advised to add the following sections to the manuscript.

a.      Significance/Novelty.

b.     Major Contributions.

c.     Problem Statement.

d.     Future Directions.

8.   The paper is found to have multiple language issues.  Therefore, thorough proofreading is required, hopefully by native English speakers. Moreover, the reference section also needs to be revised. 

Author Response

(The authors gave the same response as above.)

Round 2

Reviewer 2 Report

Authors have addressed my concerns.